# From a genome-wide screen of RNAi molecules against SARS-CoV-2 to a validated broad-spectrum and potent prophylaxis

Ohad Yogev [1,7 ✉], Omer Weissbrod[2,7], Giorgia Battistoni [1,3,7], Dario Bressan [1,3,7], Adi Naamati [1], Ilaria Falciatori[1], Ahmet Can Berkyurek[1], Roni Rasnic[2], Rhys Izuagbe [4], Myra Hosmillo[4], Shaul Ilan[2], Iris Grossman[5], Lauren McCormick[6], Christopher Cole Honeycutt[6], Timothy Johnston [6], Matthew Gagne [6], Daniel C Douek [6], Ian Goodfellow [4], Gregory James Hannon [3] & Yaniv Erlich[2]

Expanding the arsenal of prophylactic approaches against SARS-CoV-2 is of utmost importance, specifically those strategies that are resistant to antigenic drift in Spike. Here, we conducted a screen of over 16,000 RNAi triggers against the SARS-CoV-2 genome, using a massively parallel assay to identify hyper-potent siRNAs. We selected Ten candidates for in vitro validation and found five siRNAs that exhibited hyper-potent activity (IC50 < 20 pM) and strong blockade of infectivity in live-virus experiments. We further enhanced this activity by combinatorial pairing of the siRNA candidates and identified cocktails that were active against multiple types of variants of concern (VOC). We then examined over 2,000 possible mutations in the siRNA target sites by using saturation mutagenesis and confirmed broad protection of the leading cocktail against future variants. Finally, we demonstrated that intranasal administration of this siRNA cocktail effectively attenuates clinical signs and viral measures of disease in the gold-standard Syrian hamster model. Our results pave the way for the development of an additional layer of antiviral prophylaxis that is orthogonal to vaccines and monoclonal antibodies.

[1] Eleven Therapeutics, Cambridge, United Kingdom. [2] Eleven Therapeutics, Tel-Aviv, Israel. [3] CRUK Cambridge Institute, University of Cambridge, Li Ka Shing Centre, Cambridge, United Kingdom. [4] University of Cambridge, Department of Pathology, Division of Virology, Cambridge, United Kingdom. [5] Eleven Therapeutics, Cambridge, MA, USA. [6] Vaccine Research Center, National Institute of Allergy and Infectious Diseases, National Institutes of Health, Bethesda, MD, USA. [7] These authors contributed equally: Ohad Yogev, Omer Weissbrod, Giorgia Battistoni, Dario Bressan. ✉email: ohad@eleventx.com

C
ovid-19 has been one of the world's worst pandemics in modern times. While vaccines have been a major triumph, there is an urgent need to expand the arsenal of preventative measures to address some of their shortcomings[1]. First, virtually all licensed vaccines target the Spike protein[2,3], converging on a single point of failure, given the exposure to escape mutants and emerging virulent variants[4–8]. Moreover, as all monoclonal antibody (mAb) treatments target this same protein, such antigenic shifts not only hamper the protection of vaccines, but can also reduce the efficacy of a wide range of other treatment types. Second, multiple studies have shown that vaccines' protection, including against severe disease, typically wanes within just a few months, after the second[9], third[10,11], or fourth dose[12]. Third, recent lines of evidence derived from mice and non-human primates (NHPs) suggest that updated versions of vaccines exhibit diminished efficacy and may be subject to the original antigenic sin[13,14], when first exposure to a virus shapes the outcome of subsequent exposures to antigenically related strains. These data suggest the limited utility of vaccine updates for emerging variants of concern (VoCs). Forth, anti-viral drugs such as Paxlovid fell short in protecting adults from COVID-19 exposure. Finally, several studies consistently show that it is challenging to achieve high protection in immunocompromised individuals, even after repeated dosing[15], implying that the individuals who most need a vaccine are the ones least likely to benefit from it. Finally, infections in immunocompromised individuals can have a prolonged duration[16,17], which increases the risk of hyper-evolution and the emergence of VoCs, thus imposing major risks to public health.

Backed by the success of multiple previous studies where small interfering RNAs (siRNAs) were effectively used as antivirals[18–21], we envisioned that intranasally (i.n.) administered siRNAs would be particularly well suited as a vaccine-augmentation measure for infections of the upper respiratory tract, where they can be used to mitigate transmission.

To this end, we screened over 16,000 RNA interference (RNAi) triggers targeting the SARS-CoV-2 genome in order to identify hyper-potent candidates. The screen relied on a massively parallel assay, Sens.AI, which employs a synthetic biology system to recapitulate the silencing activity of each siRNA candidate against the virus. In our previous studies[22,23] we used an earlier version of Sens.AI to identify hyper-potent siRNAs against HIV and HCV. However, the previous design was inefficient, taking over 6 months to conduct. In the new design, we invented a quicker method that enhances the signal-to-noise ratio by employing statistical learning in lieu of laborious experimental steps. Extensive computational analyses and in vitro experiments yielded a cocktail of two hyper-potent siRNA candidates, which proved to be effective against all tested viral strains. Finally, intranasal administration of this siRNA cocktail was confirmed as effective in the Syrian hamster model of SARS-CoV-2.

## Results

### Screening for hyper-potent shRNA against SARS-CoV-2.
We parsed the SARS-CoV-2 genome into a series of potential short hairpin RNA (shRNA) targets (Supplemental Fig. 1). This process was conducted by tiling the genome with overlapping 50 nucleotide-long sequences, each shifted by a single nucleotide from the other. The region targeted by each shRNA comprised a stretch of 22 nucleotides positioned in the middle of the 50 nucleotide sequence, and the rest of the flanking sequence served to preserve the genomic context. We then applied multiple in silico filters to exclude target regions with low synthesis fidelity, those that do not pass a minimal threshold of conservation across viral strains, those containing sequence attributes that typically

associated with poor shRNA response, and those possessing seed regions that can potentially match a human transcript (Supplemental Table 1). In total, this process retrieved 16,471 shRNAs candidates targeting the SARS-CoV-2 genome and its sub genomic RNA1 negative strand. Finally, this library was supplemented by a set of 1,118 positive and negative control shRNAs that had been reported in previous screens against cancer-related genes in the mouse genome[22].

We synthesised these 17,589 shRNAs and their corresponding 50 nucleotide target regions using a DNA oligo pool (Twist Bioscience). Each of these oligos was 185 nucleotides long and consisted of two PCR annealing sites, the miR-30-based shRNA, a guide and its passenger strand per our design, a spacer containing cloning sites, and a 50 nucleotide region that recapitulated the target site with its genomic context (Fig. 1a). We used a series of cloning steps to introduce a Venus reporter gene to the spacer region, such that the 3'UTR of Venus included the 50 nucleotide target region and inserted this entire construct into our retro-vector library (Fig. 1b).

Our screening procedure consisted of two steps to reduce the effect of position variation that may be introduced by retro-vector integration. We first conducted the screen in a human $Dicer^{null/null}$ 293FT cell line, engineered via CAS9/CRISPR knockout (Fig. 1c; Supplemental Fig. 2a, b). The absence of $Dicer$ prevented the maturation of shRNAs, effectively uncoupling between the expression of Venus and the potency of each encoded shRNA. Overall, we transduced 1.2 million $Dicer^{null/null}$ 293FT cells with retro-vectors that encoded our library at a multiplicity of infection (MOI) of 0.8. Three days post-infection, we FACS-sorted four million Venus$^{high}$ $Dicer^{null/null}$ 293FT cells out of a total of 50 million cells. These cells represent instances of successful construct integration into genomic loci, which was reflected by adequate Venus expression. We then restored the expression of $Dicer$ by using a synthetic construct and modulated DICER expression to couple between the optical signal and the potency of each shRNA (Fig. 1d). The synthetic construct was a fusion between human $Dicer$ and a destabilising domain ($ddDicer$) that was based on a mutant human FKBP12 protein, enabling us to dial-up the activity of Dicer by using Shield-1[24]. In addition, we employed an siRNA against Dicer to induce the opposite effect, reducing its expression. The principal idea behind these various conditions was to identify a regimen in which the RNAi machinery allows hyper-potent shRNAs to inhibit their targets, but is too weak to support the activity of less potent shRNAs.

In total, we screened the library across eight different conditions of dd$Dicer$ expression (Fig. 1e). The first condition, which we assigned as $T_0$, was devoid of dd$Dicer$ and reflected the non-manipulated relative abundance of the various shRNAs. The other conditions contained increased-doses of either Shield-1 (to induce dd$Dicer$-activation) or an anti-Dicer siRNA (to inhibit Dicer). We applied each of these seven conditions to two biological replicates. We then sorted cells in each replicate into three bins based on their Venus expression: high, low, and dark, followed by sequencing of the low and dark bins in order to decrypt the level and identity of the shRNAs. We also sequenced $T_0$ unsorted shRNAs to depict the distribution of the shRNAs in the initial library. Overall, we obtained ($2_{[replicates]} \times 2_{[sorting\ bins]} \times 4_{[Dicer\ Expression\ levels]} + 1_{[T0]} =$) 17 sequencing libraries (Illumina MiSeq), each composed of 150 bp paired-end reads. In total, we obtained 36 million reads on average for each library. We parsed the 50 base-pair region that corresponded to the target from these libraries, annotated them back to their shRNA, and counted the number of unique appearances of each shRNA in each condition. Finally, we used DESeq2[25] to measure the enrichment of the shRNA in each condition versus $T_0$ and averaged the two biological replicates. This process yielded an 8-by-17,589 matrix (Fig. 1f), where each column represented an shRNA, for which each row represents one of the

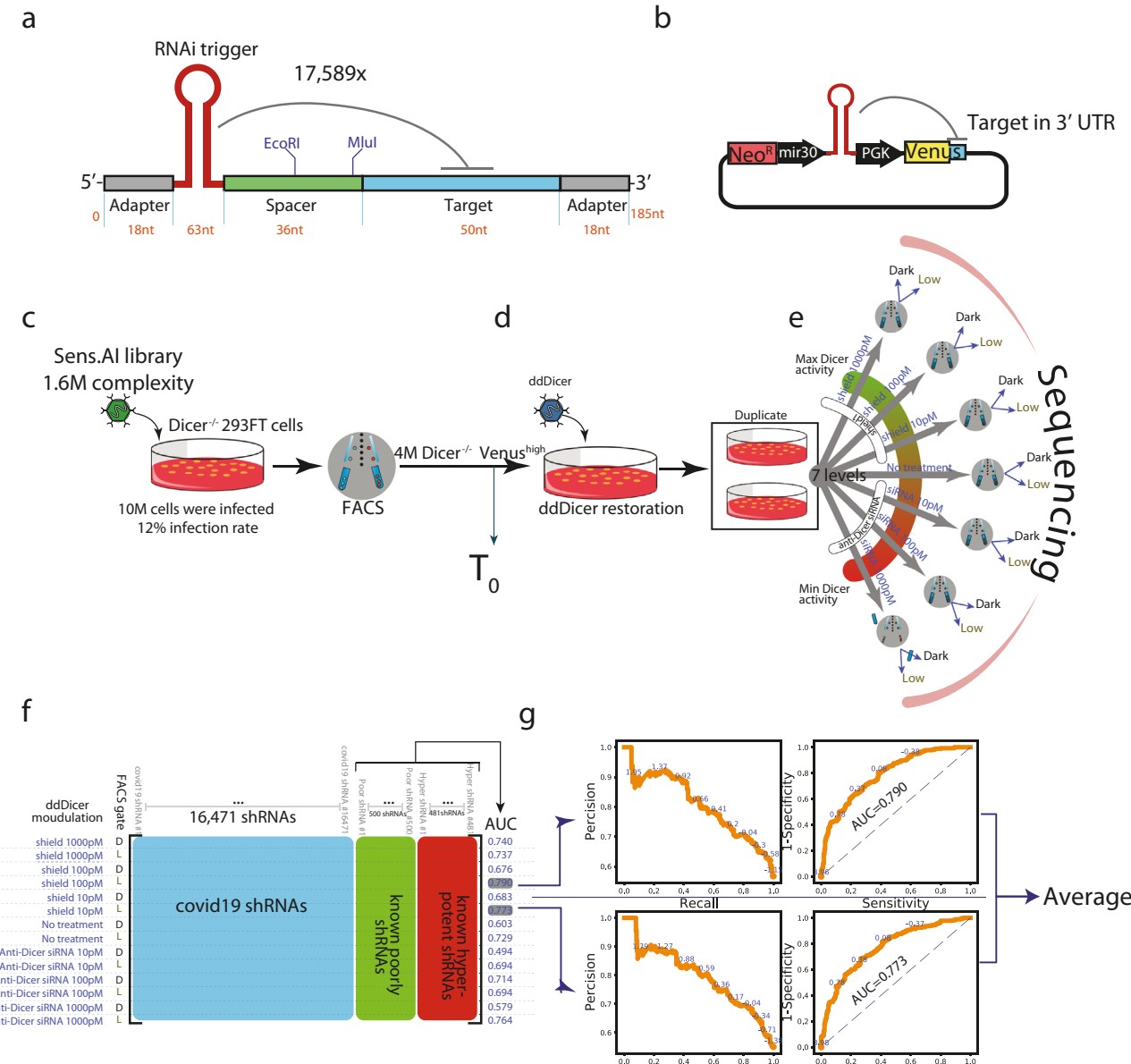

**Fig. 1 A genome-wide screen of the SARS-CoV-2 genome by Sens.AI. a** Oligo design. Each oligo had a unique RNAi trigger in the context of a miR-30 backbone along with a 50 nucleotide stretch flanking the viral RNAi target site; **b** Library design. Each plasmid contained a specific RNAi trigger and its matching target site in cis as part of the 3′UTR of a Venus reporter; **c–e** Scheme of an in vitro screen in human cells: **c** The RNAi machinery was first turned off. Dicer$^{-/-}$ 293FT cells were infected with the plasmid library and sorted for Venus$^{high}$ expression, which formed the $T_0$ read out; **d** The RNAi machinery was then turned on. We restored the RNAi machinery by ectopic expression of a destabilised Dicer (ddDicer); **e** The activity of the RNAi machinery was modulated quantitatively. Two biological replicates were subjected to seven different conditions, designed to titer Dicer expression either downward, by treatment with an anti-Dicer siRNA, or upwards, by treatment with Shield-1. Upon FACS sorting, Venus$^{low}$ and Venus$^{dark}$ cells were collected, and their oligo constructs regions were sequenced; **f** The resulting data matrix of the screen. Each row represents a specific combination of a treatment condition (the far left column) and a FACS gate (denoted as D, for Dark, and L, for Low). These row vectors describe the enrichment of each tested RNAi trigger (blue) compared to negative (green highlighted) and positive (red highlighted) controls. The area under the curve (AUC) column was calculated based on the ability to distinguish the positive from the negative controls; **g** Selection of the two best conditions. The two-row vectors with the highest AUC were selected. Precision-recall (left) and receiver operating characteristics (right) curves calculated for the intrinsic controls are shown. The screen score of each RNAi trigger was calculated as the average of these two-row vectors.

treatments (siRNA or Shield-1, each for one of two sorting bins, low and dark), and the enrichment statistic (the right-most column) was then represented by the area under the curve (AUC) as calculated by DESeq2.

Next, we used our internal controls to identify the optimal parameters that separate hyper-potent shRNAs from the rest of the library (Fig. 1f, g). We calculated the AUC for distinguishing

the poorly potent controls from the hyper-potent ones by using the DESeq2 enrichment statistic. In general, this process showed that the low Venus bins substantially outperformed the dark bins in terms of distinguishing between the hyper-potent and the poor controls. In addition, the Shield1-containing conditions performed better than the Shield-null conditions, with the 10 pM and 100 pM concentrations being the best performers. Therefore,

we decided to focus on these two conditions with the low Venus gate as the optimal conditions to distinguish between novel hyper-potent shRNAs from poorly performing ones. After focusing on RNAi triggers with high sequencing coverage, these two conditions had AUC scores approaching 80% for the internal controls. More importantly, these two conditions displayed a perfect positive predictive value (PPV) for the internal controls when restricting the recall to the top 5% of the list.

After identifying the optimal parameters, we ranked the candidate SARS-CoV-2 shRNAs using a process similar to the one employed for the internal controls. For each shRNA, we used the DESeq2 statistic of each of the conditions under the same sequencing coverage restrictions. The end result was a rank-ordered list of shRNAs across all tested conditions, with the most potent shRNA at the top, to the least potent one at the bottom.

**Validation of screen results**. Next, we validated the ranking of our screen using multiple methods. First, we analysed the correlation between the SARS-CoV-2 shRNA statistic tests across the two top-performing conditions. This analysis found the Pearson correlation between the reported statistics of the two conditions to be 72.2% ($p < 10^{-9}$), indicating that the screen results had significant internal consistency (Fig. 2a). Next, we focused on the sequence features of our shRNAs. Previous studies reported that highly potent shRNAs were typically associated with the absence of adenine in the 20th position of the guide[22]. Therefore, we evaluated the frequency of adenine as a function of the averaged screen score from the two conditions. This analysis revealed a significant correlation (Pearson = 90.4%, $p < 10^{-20}$) between the frequency of SARS-CoV-2 shRNAs without adenine in their 20th position and the average screen statistic (Fig. 2b). In fact, the top shRNAs in our screen were virtually all depleted of adenine in position 20. Finally, we compared the results of our screen to *in silico* shRNA potency predictions as produced by a published machine learning algorithm[26] (Fig. 2c). While the prediction of these algorithms was far from being perfect for each individual RNAi trigger, we found a highly significant correlation (Pearson = 90.6%, $p < 10^{-20}$) between this algorithm's scores and our average screen score.

Encouraged by these results, we manually selected ten candidates for further experimentation (Table 1; Fig. 2d). The first five candidates (S1-S3 and S5-S6) were selected mainly based on their average screen score in the two best-performing conditions while striving to represent multiple virus genes. The other five candidates (S4 and S7-S10) were also selected based on their screen scores but restricted to 22mer regions that were fully conserved between SARS-CoV and SARS-CoV-2, as we hypothesised that these regions might be applicable to future spillovers of beta-coronaviruses as well. To test our shortlisted shRNAs, we cloned their target region into the 3'UTR of mCherry and converted each shRNA to its corresponding siRNA. We then transfected each reporter with decreased doses of its corresponding siRNA to identify its half-maximal inhibitory concentration (IC$_{50}$). Eight of the ten tested siRNAs exhibited IC$_{50}$ values below 50 pM (Fig. 2e, f; Supplemental Table 2), five of which demonstrated IC$_{50}$ values below 20 pM. Only one candidate, S9, showed a relatively poor IC$_{50}$ value in this assay (IC$_{50}$ > 1.4uM). Overall, these results show that our novel genome-wide screening method identifies hyper-potent siRNAs within a single cycle.

**Discovery of siRNAs against SARS-CoV-2 using a bioinformatic pipeline**. In parallel to the Sens.AI screen, we employed a more traditional discovery pipeline to identify siRNAs against SARS-CoV-2. Our motivation was to assess the performance statistics, technical characteristics and logistical properties of our novel Sense.AI pipeline in comparison to state-of-the-art siRNA prediction algorithms. To this end, we used three open-source siRNA potency-predicting algorithms: RNAxs[27], DSIR[28], OligoWalk[29], to computationally evaluate over 815 potential target sites, focusing on regions that previously showed good results against SARS-CoV[30–32]. There was a relatively low correlation between the algorithms, at Spearman correlations of between 0.4 and 0.02 (Supplementary Fig. 3). Next, we manually selected siRNA candidates that were consistently better in all programs, excluding candidates with seed regions complementary to the human transcriptome. This list yielded 88 siRNAs that we synthesised and tested by the same reporter assay (at 1 nM per candidate) described above. We then prioritised the most promising 27 siRNAs and retested them at 500 pM and 100 pM concentrations (Supplementary Fig. 4), finding that 9 of these 27 candidates inhibited the reporter expression by more than 50% (100 pM).

**siRNAs conferring protection against multiple SARS-CoV-2 variants**. Next, we assessed our top candidates from the sensor screen and from the bioinformatic pipeline using a gold standard live SARS-CoV-2 in vitro infection assay. We transfected Vero E6 cells with 100 nM of each of the Sens.AI siRNA candidates in triplicates. As a negative control, we also transfected cells with a mock siRNA that targeted eGFP. After 24 hours, we challenged the cells with 60xTCID$_{50}$, 600xTCID$_{50}$, or 6000xTCID$_{50}$ of the live SARS-CoV-2 (ancestral strain). Finally, we measured the level of viral load 48 hours after the initial infection via qPCR, probing the RdRP and the E genes.

We found that six out of the nine tested siRNAs from our screen were able to dramatically lower the amount of viral RNA. While the results were qualitatively consistent across all three virus titers tested (Fig. 3 and Supplementary Fig. 5a, b), we decided to focus on the 600xTCID$_{50}$ titer for future experimentations, because it yielded the greatest dynamic range (Fig. 3a). In these conditions, our best performing five siRNAs repressed genomic viral load by >95%. Interestingly, both S8 and S10 showed weak responses, at the level of ~10% SARS-CoV-2 inhibition compared to the control siRNA, despite very high potency in the reporter assay (IC50 < 20 pM). However, unlike the other siRNA candidates, these two target the virus's negative strand, which is an intermediary in the replication process. Therefore, we hypothesised that targeting this intermediary RNA molecule likely would not interfere with viral replication. To further confirm our top candidates, we assessed the effect on live virus infectivity in cells using TCID50 assay (Fig. 3b). Again, we found a strong inhibitory activity of the top three out of four siRNAs, with viral repression exhibiting approximately two orders of magnitude compared to the GFP siRNA control. We next used the same settings to test five of the most potent siRNAs from the open-source discovery method (Supplementary Table 3). Only one candidate, Hel14, displayed repression levels of the viral load by >95%, on par with the levels exhibited by the top siRNAs from our Sens.AI screen (~90% inhibition) (Fig. 3c).

Next, we searched for the optimal combination of siRNA pairs out of our best-performing candidates. These cocktails included four siRNAs from the Sens.AI screen, augmented by three of the most promising siRNAs from the open-source discovery pipeline. We tested 14 different 2-siRNA cocktails in the live virus assay and compared the results to repression by S3 alone, because as a monotherapy it had shown the highest repression level (Fig. 3d). We identified five cocktails where the two siRNA components exhibited a synergistic effect, multiplied by several folds in comparison to repression by S3 alone at the same concentration. S5 turned out to be a repeat component in most of these cocktails

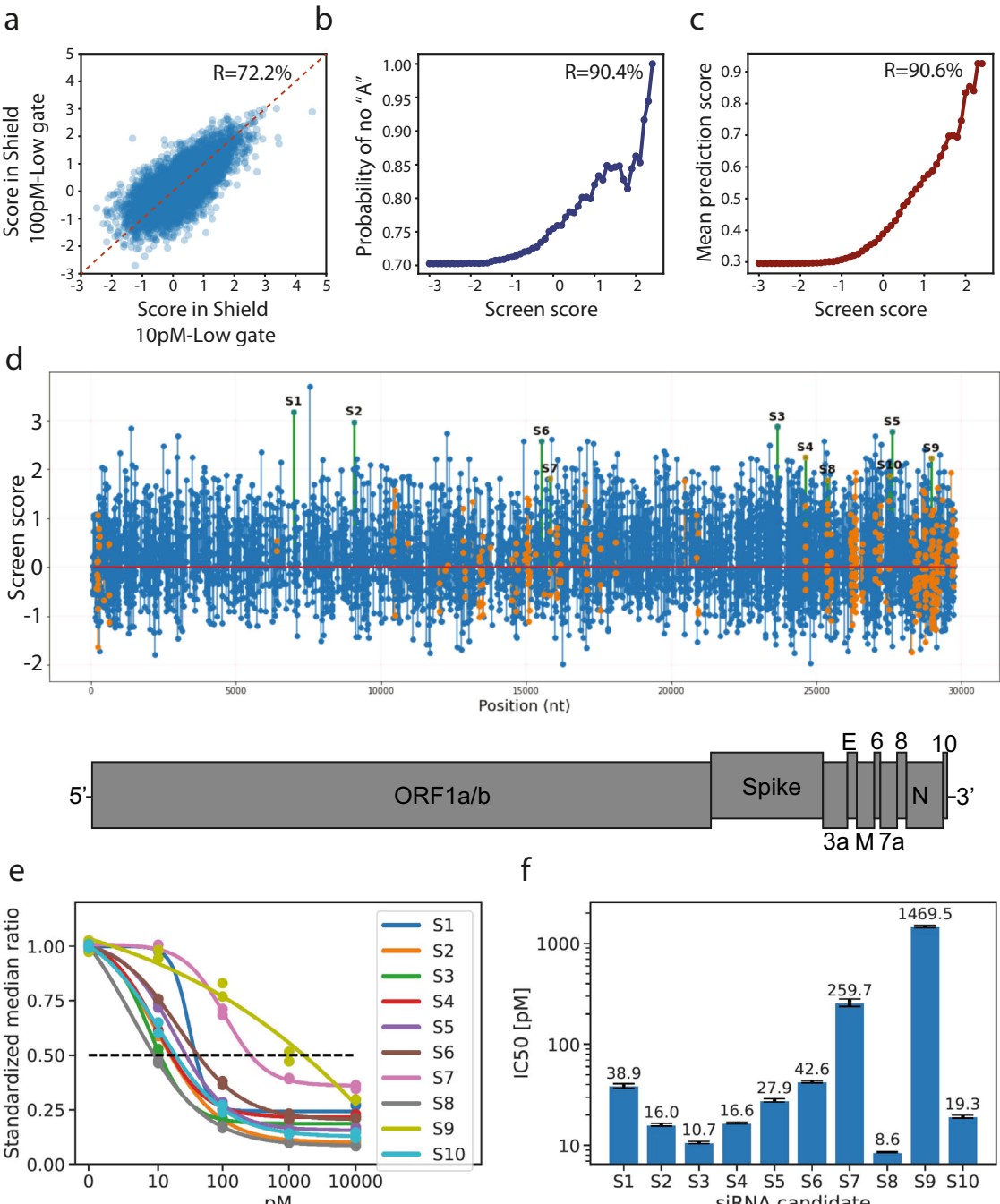

**Fig. 2 Validation of screen results for $n = 17{,}589$ shRNAs. a** Screen scores had a high internal consistency. The screen results showed a Pearson correlation of 72.2% between the scores in the two top-performing conditions; **b** Enrichment of features associated with potent shRNAs. The probability of *not* finding an adenine in position 20 of RNAi triggers targeting SARS-CoV-2 increased with screen scores, recapitulating previous studies indicating that when adenine occupies position 20 it weakens shRNA maturation; **c** Screen scores highly correlated with bioinformatic predictions. The screen scores, representing enrichment statistics as calculated by DESeq2, were highly correlated with a published machine learning algorithm predicting the potency of shRNAs; **d** Screen results and the selected candidates. The graph shows the resultant screen scores of each RNAi trigger that passed the quality threshold, as a function of position along the viral genome. Orange: RNAi triggers that targeted conserved regions between SARS-CoV and SARS-COV-2. Green: The selected ten candidates. Blue: all other RNAi riggers; **e** Dose-response curves of the selected ten candidates. siRNAs were tested using a reporter assay. The plot represents the standardised median ratio between the expression of mCherry (reporter gene) and GFP (control gene); **f** Potency scores of each of the 10 selected candidates. The $IC_{50}$ value of each of the 10 candidates was calculated based on the dose-response curves in (**e**). Standard errors for IC50 computations were computed using statistical bootstrap over all data points at each concentration level.

and we decided to prioritise two of these cocktails moving forward: S5/S3 and S5/Hel14.

To further validate these siRNA cocktails, we tested their activity in an Air-Liquid interface culture which mimics better the respiratory tract compared to the VeroE6 cells. We transfected the ALI culture from the apical side with 100 nM of each cocktail. As a negative control, we also transfected cells with a mock siRNA that targeted eGFP. After 24 h, we challenged the cells

**Table 1 The ten shRNAs identified by the Sens.AI screen that were selected for further validations and development.**

| Label | Strand | Gene | Position | Screen score | SARS-CoV+ | IC50 [pM] | RNAi trigger guide | siRNA sequence |
|---|---|---|---|---|---|---|---|---|
| S1 | + | nsp3 | 6989 | 3.18 | No | 38.9 | TTAAAACACCTAAAGCAGCGGT | UUAAAACACCUAAAGCAGCGGUTT |
| S2 | + | nsp4 | 9067 | 2.97 | No | 16 | TTCATAAGCAACAGAACCTTCT | UUCAUAAGCAACAGAACCUUCUTT |
| S3 | + | Spike | 23646 | 2.87 | No | 10.7 | TAAGCAACTGAATTTTCTGCAC | UAAGCAACUGAAUUUUCUGCACTT |
| S4 | + | Spike | 24610 | 2.24 | Yes | 16.6 | TTTAGTAGCAGCAAGATTAGCA | UUUAGUAGCAGCAAGAUUAGCATT |
| S5 | + | ORF7A | 27611 | 2.77 | No | 27.9 | TTAGGTGAAACTGATCTGGCAC | UUAGGUGAAACUGAUCUGGCACTT |
| S6 | + | RdRP | 15528 | 2.58 | No | 42.6 | TAAAAGTGCATTAACATTGGCC | UAAAAGUGCAUUAACAUUGGCCTT |
| S7 | - | RdRP | 15819 | 1.82 | Yes | 259.7 | AAGGTCAGTCTCAGTCCAACAT | AAGGUCAGUCUCAGUCCAACAUTT |
| S8 | - | Spike | 4464 | 1.79 | Yes | 8.6 | TACATTACACATAAACGAAACTT | UACAUUACACAUAAACGAAACUTT |
| S9 | + | N | 28969 | 2.23 | Yes | 1469.5 | TTGGCCTTGTTGTTGTTGGCCT | UUGGCCUUGUUGUUGUUGGCCUTT |
| S10 | - | ORF7A | 2324 | 1.87 | Yes | 19.3 | TACGAGGGCAATTCACCATTTC | UACGAGGGCAAUUCACCAUUUCTT |

+Indicates whether the target site is conserved in SARS-CoV, in addition to SARS-CoV-2.

from the apcical side with 600xTCID$_{50}$ of the live SARS-CoV-2 (ancestral strain). Finally, we measured the level of viral load at 48, 72 and 96 hours after the initial infection via qPCR, probing the RdRP and the E genes. We found strong inhibition of viral replication for both cocktails under these conditions (Fig. 3e).

Our cocktails proved to be highly resistant to emerging VoCs. First, we tested the cocktails against the ancestral strain versus the Delta variant. The cocktails conferred substantial repression against Delta, on par with their effect on the ancestral strain (>95% repression) (Fig. 3f). Interestingly, S5 was tolerant and showed efficacy against the Delta strain despite the fact that it possesses a mutation in position 14 of its target site. Second, we tested the S5/Hel14 cocktail against Omicron BA.1 using a similar setting. Similar to other reports[33], the Omicron variant did not replicate as fast as other VoCs in vitro. Since these lower replication rates reduced the dynamic range of our assays, we added two positive controls, chloroquine and molnupiravir, both of which demonstrated as potent inhibitors of Omicron BA.1. The activity of the S5/Hel14 cocktail was indeed diminished in the Omicron variant. Nevertheless, it inhibited viral replication to a similar level as was induced by the positive control treatments (Fig. 3g). This finding suggested that the more modest inhibition was likely the result of a lower dynamic range due to slow viral replication, rather than due to reduced potency of the RNAi cocktail itself. Finally, we tested the activity of the cocktails against our novel replicon system, which recapitulated the function, but not infectivity of the Beta SARS-CoV-2 strain[34]. Consistently, the cocktails repressed the replicon by 10-15 fold, indicating that they confer a robust inhibition profile across diverse VoCs[34]. Importantly, analysis of high throughput sequencing data showed that cells that were treated with the S3/S5 cocktail exhibited a specific profile of depletion of reads at the siRNA cleavage sites (Fig. 3h). These data provide mechanistic support to confirm that the observed reduction in viral load was directly related to siRNA silencing.

**High throughput saturation mutagenesis to assess siRNA cross-reactivity.** Next, we explored the cross-reactivity of our RNAi strategy against future VoCs, given its intended use in the context of pandemic preparedness. To this end, we developed a saturation mutagenesis assay using our Sens.AI strategy. We repeated the same RNAi-off/RNAi-on strategy as the SARS-CoV-2 screen. The only difference was that in the previous screen we employed a variety of shRNA triggers and a perfectly matched target site, whereas in this assay, we fixed the shRNA trigger to be of S5 and created a series of mutated target sites (Fig. 4a). We used this strategy to screen 2,143 mutations in the S5 target site, exhaustively evaluating virtually every possible one and/or two substitutions in this site. To the best of our knowledge, this is the largest in-cellulo saturation mutagenesis ever conducted with an RNAi trigger.

We validated the saturation mutagenesis screen by replicating previous trends about siRNA target mismatches. We stratified the results based on the number of mismatches. On average, a single mismatch in the target site reduced Sens.AI scores by 6% compared to no mismatch. As expected, this figure was significantly smaller (t-test, $p < 3 \times 10^{-10}$) than that observed for double mismatches, which reduced the Sens.AI scores by 31% on average. Next, we analysed the effect of the position of a single mismatch on the activity of S5 (Fig. 4b). Similarly, consistent with previous studies[35], the seed region showed the greatest sensitivity to mismatches, whereas the cleavage site displayed a relatively smaller sensitivity to these mutations.

Overall, our high throughput screen showed that the S5 target site could tolerate a wide variety of mutations without a

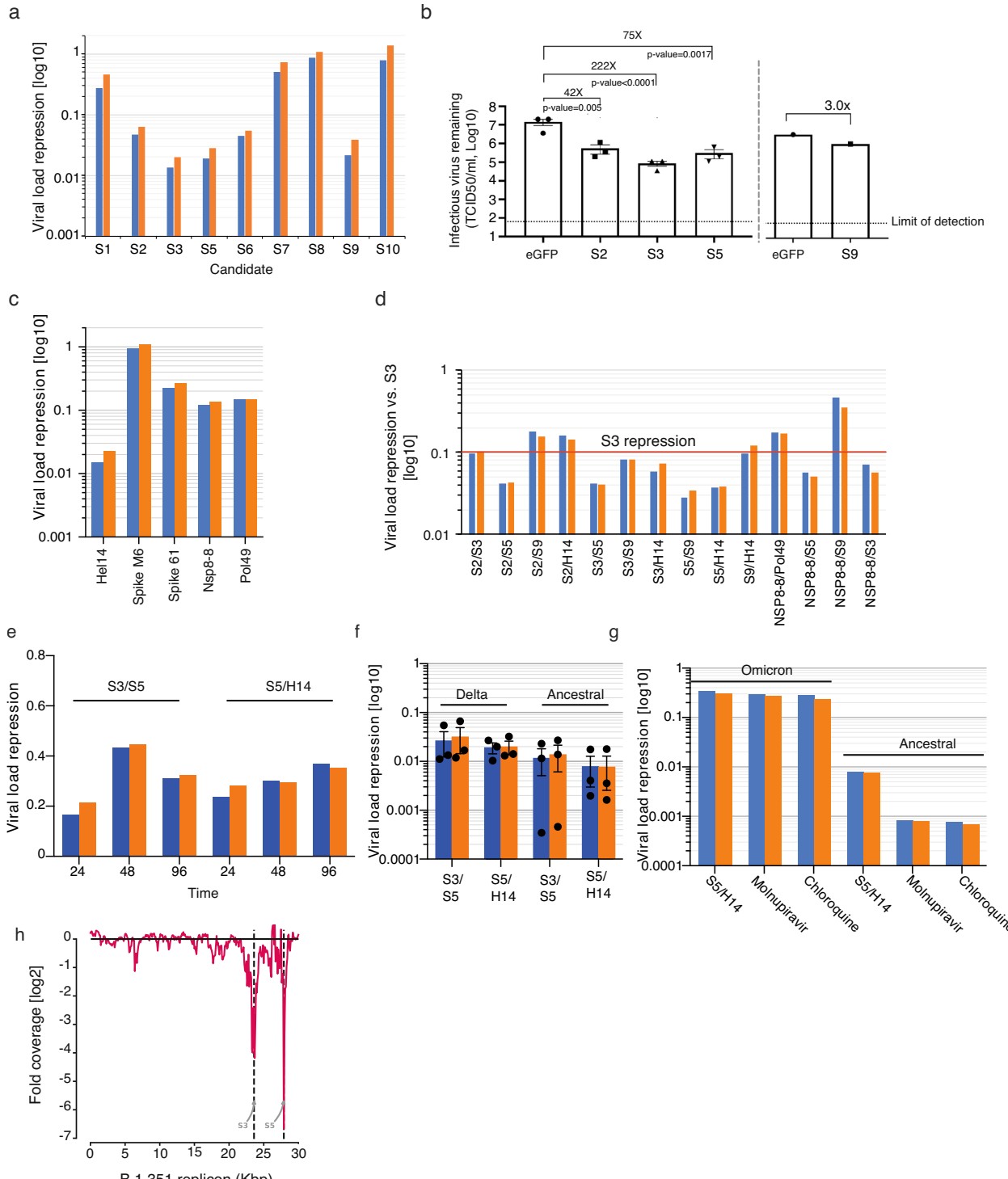

**Fig. 3 siRNAs repress live SARS-CoV-2 replication in VeroE6 cells.** Blue and orange: qPCR results of the E and RdRP transcripts, respectively. Unless noted otherwise, the 100% viral load was calibrated to viral level after treatment with an anti-GFP siRNA. **a** Viral load of SARS-CoV-2 (ancestral strain) after treatment with the top siRNA candidates from the Sens.AI screen; **b** $TCID_{50}$ levels of SARS-CoV-2 (ancestral strain) after treatment with four of the top siRNA molecules. In each batch of the experiments, an siRNA against eGFP was used as a negative control (left panel $n = 3$, right panel $n = 1$, error bars=SEM, significance was calculated using t-test); **c** Viral load of SARS-CoV-2 (ancestral strain) after treatment with the top siRNA from the bioinformatic pipeline; **d** The effect of various siRNA cocktails against SARS-CoV-2 (ancestral strain). The results were calibrated to the repression of S3 at the same concentration; **e** Viral load of SARS-CoV-2 (ancestral strain) after treatment with siRNA cocktails in ALI culture. Viral load was measured in media collected from the media at 48, 72 and 96 hours postinfection ($n = 2$ independent samples, error bars=SEM). **f** Viral load post-treatment with siRNA cocktails against SARS-CoV-2 Delta versus the ancestral strain ($n = 3$ independent samples, error bars=SEM). **g** Viral load of SARS-CoV-2 Omicron versus the ancestral strain after treatment with the S5/Hel14 cocktail and other types of antivirals; (**h**) DeSEQ2 analysis of SARS-CoV-2 replicon treatment with the S3/S5 siRNA cocktail. We observed a sharp coverage decrease around the S3 (~23.5kbase) and S5 (~28kbase) cleavage sites along the replicon sequence (FDR values $4 \times 10^{-83}$ and $6 \times 10^{-22}$, respectively). Error bars in panels **e-f** represent standard diviation, based on three replicates.

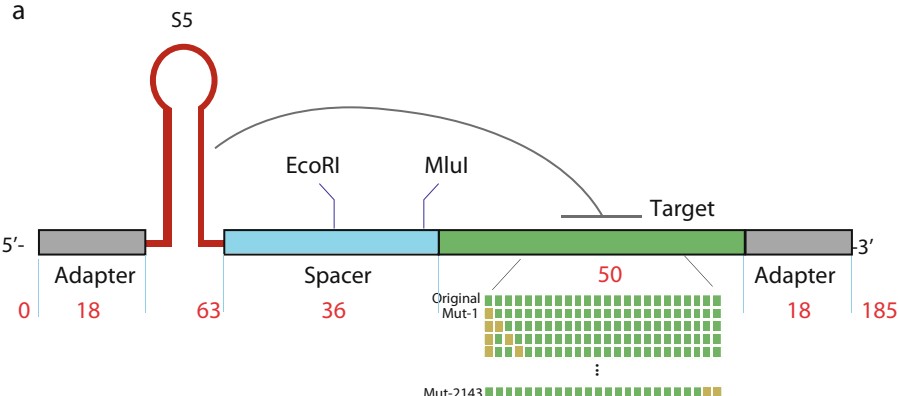

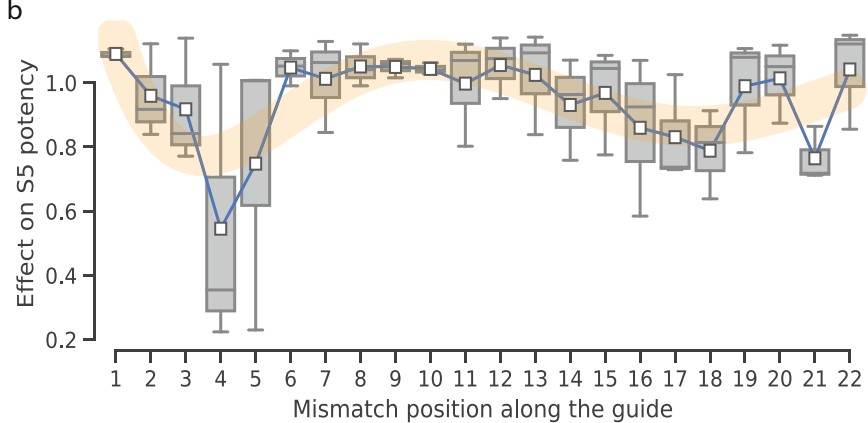

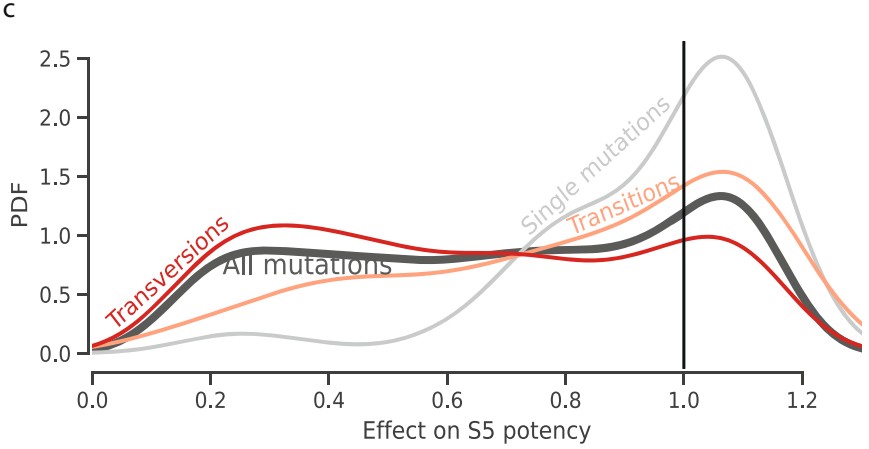

**Fig. 4 Saturation mutagenesis. a** The design scheme of oligos used in this setting. The library consisted of the S5 as the shRNA trigger with 2,143 mutations, exhaustively depicting every possible single- and double-mismatch in the siRNA target site; **b** The effect of mismatches stratified by position for $n = 66$ shRNAs with a single mismatch. The X axis represents positions along the guide strand. Blue: mean values. Yellow: smoothed mean effect; Squares represent the mean, horizontal lines represent the median, box edges represent the 25% and 75% quartiles, and the whiskers represent the furthest data points within up to 50% of the interquartile range. **c** The distribution of the effect of mutations on the activity of S5. The vertical line at 1.0 (X axis) signifies scores that are identical to the screen score of a target site devoid of any mutation. Black: the distribution of effects of all 2143 single and double mutations. Grey: the distribution of all single mutations. Orange: the distribution of all double transitions. Red: the distribution of all double transversions.

significant loss in potency (Fig. 4c; Supplementary Fig. 6). Most single mutations (56%) enhanced the Sens.AI score and were expected to increase S5 potency, which could be attributed to Ago2 dissociation dynamics[35]. In contrast, we estimated that S5

would lose most of its potency in the case of a double mutation in the target site. To better understand the dynamics of these double mutations, we compared the estimated effect of double transitions to double transversions. Our results show that double transitions

were much better tolerated. Most double transitions resulted in <32% reduction in the Sens.AI score, whereas most double transversions resulted in >51% reduction in this score. Based on previous studies, the former mutation type is seven times more common than the latter[36], suggesting that S5 could tolerate to some extent the more prevalent type of double mutations in the target site.

**In vivo validation**. In order to assess the preventative efficacy of our siRNA cocktails in a disease model of COVID-19, we administered the S5/Hel14 cocktail to Syrian hamsters as a pre-emptive measure to a live virus challenge. We decided to use this cocktail over the S3/S5 one since S3 targets the Spike-encoding region, which is prone to mutations as a vaccine- and mAb-escape mechanism. We employed a dosing regimen that consisted of one i.n. dose of ~400ug/kg of the siRNA cocktail per day on days −7, −3, and −1 using our proprietary lung-selective delivery formulation. In addition, we used the same dosing regimen to treat another group of hamsters with a known and highly potent siRNA against hepatitis C virus (HCV), as a negative control. As there is no available intranasal drug we could use as a positive control, we administered bamlanivimab[37] (LY-COV555, which received an emergency use authorization by the FDA for COVID-19 treatment) intraperitoneally (i.p.) on day −1 to the positive control group. Each group was composed of six male hamsters and we challenged them intranasally at day 0 with $4 \times 10^3$ PFUs of the WA-1 ancestral SARS-CoV-2 strain.

Our results show that the siRNA cocktail conferred protection against SARS-CoV-2 infection when used as pre-exposure prophylaxis (PrEP) in the gold-standard Syrian hamster disease model (Fig. 5a). The negative control group exhibited an average of >7% weight loss at Day 5 postinfection, consistent with

previous studies[38–40]. The siRNA-treated group benefited from significant protection against weight loss compared to this negative control group ($p < 0.05$; bootstrap hypothesis testing and Bonferroni adjusted) (Fig. 5b). This weight loss was statistically indistinguishable from the positive control group that was treated with bamlanivimab. In addition, we observed an order of magnitude suppression of viral load in the lungs of the active treatment group (Day 5) compared to the negative control, as measured by qPCR measurement of the RdRP gene ($\Delta VL_{lung} = 10.3x$, $p_{lung} < 0.001$; bootstrap hypothesis testing and Bonferroni adjusted) (Fig. 5c). Similarly, we also observed a significant suppression, albeit to a lesser extent, of viral load in the nares ($\Delta VL_{lung} = 2.5x$, $p_{lung} < 0.05$; bootstrap hypothesis testing and Bonferroni adjusted) (Fig. 5d). In both cases, the antibody was more effective than the siRNA treatment, suggesting that additional dose optimization studies are required in order to fully develop the siRNA cocktail into a viable prophylactic.

We also tested the same siRNA cocktail in three additional formats: chemically modified siRNAs administered intranasally (variation #1; Supplemental Fig. 7), same chemically modified siRNAs administered by nebulization (variation #2), and naked siRNAs administered by nebulization (variation #3) (Methods). Both treatment variation #1 and #2 showed significant reduction in the lung's viral load ($p_{lung\text{-}variation\ \#1} < 0.001$; $p_{lung\text{-}variation\ \#2} < 0.001$; bootstrap hypothesis testing and Bonferroni adjusted) (Supplemental Fig. 8a). In addition, we found a smaller but significant reduction of in nares' viral load ($p_{lung\text{-}variation\ \#1} < 0.001$; $p_{lung\text{-}variation\ \#2} < 0.05$; bootstrap hypothesis testing and Bonferroni adjusted) (Supplemental Fig. 8b). However, none of these variations protected against weight loss as measured on Day 5 (Supplemental Fig. 8c), suggesting that they may display slower pharmacokinetics than the primary treatment arm, or induce a tolerability challenge.

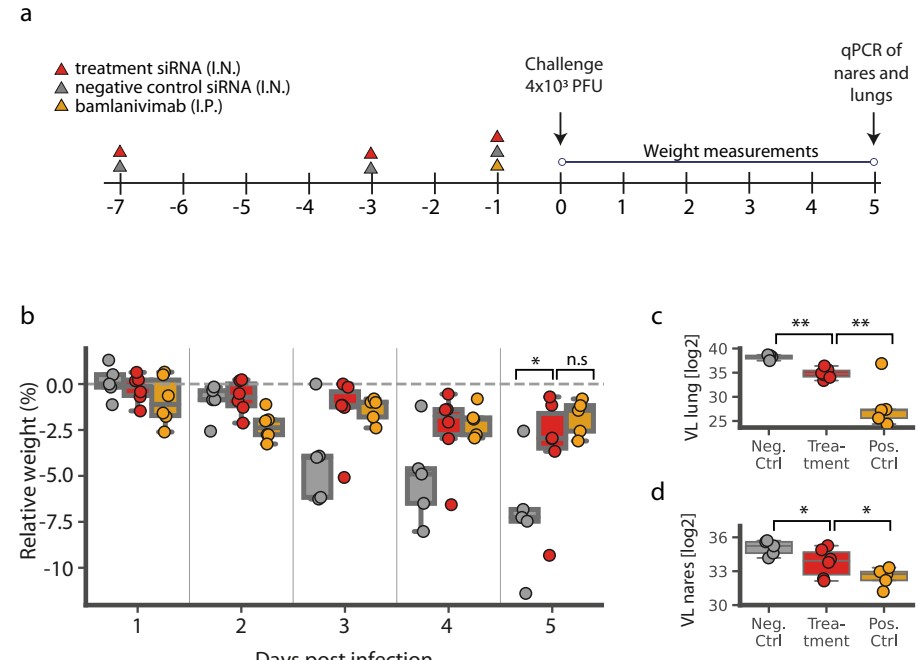

**Fig. 5 Prophylactic treatment against SARS-CoV-2 infection in Syrian hamsters. a** Dosing regimen. Syrian hamsters ($n = 6$, per group) pre-treated with a non-targeting siRNA (negative control, grey), the LY-CoV555 antibody (positive control, yellow) or our lead siRNA cocktail (treatment, red) were infected with $4 \times 10^3$ PFUs of the ancestral SARS-CoV-2 strain; **b** Weight change post infection. Box plot of the change in weight by treatment group relative to time postinfection; **c, d** Viral load at day 5 postinfection. All measurements were based on qPCR of the RdRP gene five days postinfection from homogenised lungs **c** and nares **d**. In all panels $p$ value is presented as: *<0.05,**<0.001. Ctrl: control; VL: viral load; Neg.: negative; Pos.: positive; tment: treatment; n.s.: not significant. In panels **b–d**, horizontal lines represent the median, box edges represent the 25% and 75% quartiles, and the whiskers represent the furthest data points within up to 50% of the interquartile range. P-values were computed using parametric bootstrap (Methods).

Taken together, our results show that siRNA treatment can effectively protect the upper respiratory tract against SARS-CoV-2 infection, significantly attenuating infection as reflected by measurements of viral load and clinical manifestations prototypically exemplified by weight loss.

## Discussion

Prophylaxis, particularly the kind resistant to the emergence of novel variants of concern (VoCs), has been repeatedly identified as a missing component in the arsenal of therapies used to fight the continuous COVID-19 pandemic. siRNAs present an attractive modality for prophylactic treatment, and indeed during the first SARS outbreak, different researchers have identified siRNAs to target the virus genome. Nevertheless, out of more than 9000 published sequences, only 12 siRNAs had a perfect match to the SARS-Cov2 genome. In this study, we report, for the first time to our knowledge, on a systematic, genome-wide RNAi screen against SARS-CoV-2. We tested over 16,000 RNAi triggers in a massively parallel reporter assay and validated the best performers in an in vitro live virus assay. In addition, we tested 88 siRNAs identified via open-source, in-silico discovery methods. We then tested multiple siRNA pairs as cocktail treatments in the live virus assay in order to identify instances where the siRNA components confer synergistic effects when combined. These cocktails proved to be active against three different strains of the virus, namely Beta, Delta and the ancestral one. An exhaustive screen of all 2,143 single and double mutations in one of the target sites further confirmed the low likelihood of loss of activity against future VoCs. Finally, we showed the efficacy of these cocktails as pre-exposure preventatives against SARS-CoV-2 infection in Syrian hamsters.

Our study also has certain limitations. First, in their current constellation, the siRNA cocktails were designed as acute prophylactics. It is unclear for how long they remain effective and thus might be most appropriate in situations where high exposure risk is present, such as in the case of front-line workers, immunodepression (transient or chronic), an outbreak in the household, and during the height or an infection "wave". Second, while our siRNA cocktail protected Syrian hamsters from disease, the effect was mainly observed in the upper respiratory tract. While this is the chief site of infection and shows great promise in terms of mitigating transmission, it will require further adaptation for use in a treatment setting, which would require optimization for delivery to the lower respiratory tract and through to the lung parenchyma. Third, for the purpose of this proof-of-principle study, we adopted a pre-exposure treatment regimen that was composed of multiple days' administration prior to infection. We aim to further optimise the regimen prior to advancing into NHPs and ultimately to clinical trials in humans.

Despite these caveats, the current study substantially contributes to the global efforts to curb COVID-19, as well as future other viral pandemics of similar mortality and morbidity proportions. Herein we illustrate the efficacy of intranasally administered siRNA cocktails as pre-exposure preventatives resilient to the likely mutational evolution of culprit pathogens. This is particularly encouraging given the alarming rate at which we are currently witnessing the emergence of novel SARS-CoV-2 variants resistant to therapies of the neutralising mAb and vaccine classes. Importantly, our proposed approach is not affected by Spike mutations, nor does it rely on a functional immune system. Instead, it harnesses the natural RNAi pathway, active in the lining epithelial cells of the respiratory tract, which are the focal point of infection transmission, and is completely orthogonal to vaccines and other immuno-modulatory approaches. Finally, the global proportions of the current pandemic, and the restrictive cold-chain storage and shipment requirements associated with many of the vaccine and mAb-approved therapies urgently call for a durable solution that would also be accessible to difficult-to-reach rural, less affluent populations. Our intranasal administration allows for widespread deployment that is not reliant on novel or advanced medical infrastructure for accessibility. We believe that this study illustrates a paradigm shift in the approach to pandemic preparedness, and we expect that our discovery and validation pipeline will be applicable to other emerging pathogenic threats, including novel beta-coronaviruses and influenza.

## Methods

**Cell cultures.** HEK293FT (ThermoFisher Scientific) and VeroE6 (ATCC) cells (and their derivatives) were grown in DMEM (Dulbecco's Modified Eagle Medium), supplemented with 10% Foetal Bovine Serum, 100 U/ml penicillin and 100 μg/ml streptomycin (all Thermo Fisher Scientific), at 37 °C with 5% CO2. The HEK293FT cell-line was obtained from Thermo Scientific (cat. No R70007). Air-Liquide Interface culture (MucilAir) was purchased from Epithelix.

**Generation of a DICER Knock-Out cell line**. The HEK293FT-Dicer Knock-Out (KO) cell line was engineered by CRISPR/Cas9 in the parental 293FT cell-line. The guide RNA (gRNA) had the following sequence: AAGAGCUGUCCUAUCA-GAUC. The gRNA was modified with a phosphorothioate modification to prevent nuclease degradation and was obtained from Merck-Millipore. 80 pmol of gRNA were complexed with 4 ug of TrueCut Cas9 Protein (ThermoFisher) and transfected to the cells using the Amaxa nucleofector. KO efficiency was determined using Sanger sequencing.

**Reporter assay experiments**. For each candidate siRNA we extracted its target sequence within a 150 base-pair context. The target sequence was cloned into pLMN-ZsGreen-Neomycin (Transomics) at the 3'UTR of a constitutively expressed mCherry reporter protein. HEK293FT cells were transfected with Lipofectamine 3000 (Life Technologies) according to the manufacturer's guidelines, either with the sensor alone or with the sensor plus siRNA at the relevant concentration. A second plasmid expressing eGFP was co-transfected at a 1:1 molar ratio, serving as an internal control for transfection efficiency. Cells were collected and analysed 48 h post-transfection using a MACSquant VYB flow cytometer (Milteyi Biotec). The potency of each candidate was evaluated as the median ratio of mCherry to eGFP radiance amplitude. Supplementary figure 9 present the gating strategy used in this experiments.

**Cloning of the sensor library**. The shRNA-Sensor library was assembled via a two-step procedure. A library of ~20,000 oligonucleotides in which each shRNA was joined to its cognate Sensor by a linker harbouring EcoRI and MluI restriction sites was obtained from Twist Bioscience. The library was first PCR-amplified and cloned using XhoI and MfeI in a recipient retroviral vector. The latter contained the Hygromycin-resistant miR30, including its 5' portion (5'mir30). In the second step, a 3'mir30-PGK-Venus cassette was inserted between the shRNA and its Sensor, integrated via the EcoRI and MluI sites in the linker.

**Plasmid cloning**. All plasmids for the reporter assay and destabilised Dicer were built on a lentiviral pZIP scaffold obtained from Transomics (pZIP-SFFV-ZsGreen-Puro), which had been previously modified to switch the SFFV-ZsGreen cassette with EF1alpha promoter alone. For the reporter assay, we cloned an expression cassette downstream of the EF1aplha promoter via Gibson Cloning, which consisted either of a 3xFLAG-EGFP-NLS or mCherry-STOP-150bp sensor fragment. For the destabilized ddDicer, the destabilization domain[41] and human Dicer1[42] were amplified via PCR and assembled in pZIP-EF1aplha via a 3 way Gibson cloning. The final construct had the following structure EF1a::dd-Dicer1-IRES-Puro.

**Live virus experiments**. All in vitro studies with a live virus were conducted in containment level 3 facility at the Cambridge Institute Therapeutic Immunology and Infectious Disease (CITIID) under approved standard operating procedures and protocols.

Clinical isolate of Sars-cov-2/human/Liverpool/REMRQ001/2020, a first-wave isolate designated as WT, was propagated in Vero-E6 and primarily used for live virus infections in this study. Additional experiments were conducted using newly emerged Delta and Omicron variants. Propagated live infectious SARS-CoV-2 viruses, B.1.1.617.2 (Delta) and B.1.1.529 (Omicron) were kindly received from Professor Wendy Barclay (Imperial College London) and Dr Jonathan Brown (Imperial College London) as part of the work conducted by G2P-UK National Virology Consortium. The virus isolate matching Omicron variant, kindly donated by Gavin Screaton at Oxford University, was propagated on Vero-ACE2-TMPRSS2 (VAT) cells for 3 days until cytopathic effect was observed.

Transfected Vero E6 cells with control GFP or experimental siRNA/s were prepared in 96-well. Unless indicated differently, 10,000 VeroE6 cells were transfected with 100 nM of siRNA treatment and infected with SARS-Cov-2 24 hours post-siRNA treatment. Cells were infected in biological triplicates with either SARS-CoV-2 WT, Delta and Omicron variant at m.o.i. 1 or 0.1 $TCID_{50}$ per cell in 50 µl DMEM with or without known SARS-CoV-2 inhibitors. Infections were incubated up to 48/72 h postinfection (p.i), and cell culture supernatants were harvested at 0, 24, 48 and/or 72 h p.i. where viral RNAs were quantitated by RT-qPCR and/or infectious virus units were titrated by $TCID_{50}$. After that, 40ul of the media was collected directly into 160ul of TRIzol™ LS Reagent (ThermoFisher). Viral RNA was extracted using the Direct-zol-96 RNA Kits (ZYMO research) and used as a template for qRTPCR to measure viral copy number. Chloroquine Diphosphate (BioVision) was used at a final concentration of 50uM and Molnupiravir (Focus Biomolecules) was used at a final concentration of 20uM. The sequencing of the Beta replicon was described previously[34].

**TCID50 assay**. Ten-fold serial dilutions of collected virus supernatants were prepared in DMEM culture media. Of these dilutions, 50 µl was inoculated onto monolayers of Vero E6 cells grown on 96-well plates and incubated at 37 °C in a 5 % $CO_2$ incubator. Virus titres were collected at 4 days p.i. and expressed as $TCID_{50}$ ml$^{-1}$ values by the Reed–Muench method[43].

**SARS-CoV-2 detection**. Viral genome copy number was measured by qPCR using the Charité/Berlin Primer Probe Panel (IDT) and the TaqPath™ 1-Step Multiplex Master Mix (ThemoFisher). We then compared the average Ct values of each condition to the Ct values of eGFP siRNA.

**Adaptation of the screen to investigate siRNA mismatch tolerance**. The oligos for the S5 mutagenesis saturation assay were obtained from Twist Bioscience and cloned into the shRNA-Sensor library as described above. Similar to the previous screen, we also incorporated 948 control shRNA from Fellmann et al.[22], 511 which are highly potent and 437 of low potency. This screen followed a similar structure to the genome-wide screen with a few modifications: (a) we modulated the RNAi machinery with only three conditions: upregulation of Dicer, downregulation of Dicer, and no modulation. A machine learning pipeline was built to distinguish between the potent and weak control shRNA. The best classifier was chosen using cross-validation, and had an 81.1% precision rate on average (s.e. of 1.14%). The classifier assigns a potency score for each trigger-target pair in the assay. The effect of each mutated target was determined as the classifier score for the mutated trigger-target pair over the score of the perfectly matched trigger-target pair.

**In vivo experiments**. Male Golden Syrian hamsters age 6–8 week were treated with siRNA cocktails at days -7, -3 and -1 pre-infection. Intranasal (IN) administration was performed on hamsters in Study using a proprietary formulation. Each hamster was placed on a sterile surgical pad and lightly stretched out to better place a firm grip on the scruff. The hamster was turned on its back to allow the hamster to breathe and be comfortable. With the neck and chin flat and parallel to the pad, the tip of the pipettor was placed near the left nostril of the hamster at a 45-degree angle, and 5 µL of dosing material was administered to the nostril with a 2–3 sec interval in between for a total of 25 µL/nostril. The hamster was held in this position for 5 seconds or until it regained consciousness, then the administration was repeated for the other nostril for a total of 50 µL/hamster. After the procedure, the hamster was returned to its cage and monitored for 5–10 minutes for any adverse reactions.

For nebulizer administrations siRNA were diluted in PBS + Gelatin 0.5 mg/ml. Aerosol was produced using Vibrating Mash Nebulization (VMN), and nebulization was performed in a Biological Safety Cabinet (BSC).

When indicated, 1 mg of mAb555 was administered Intravenously at day -1. At day 0 hamsters were infected with SARS-CoV-2 using intranasal infection of $4 \times 10^3$ PFU virus. At day 5 postinfection, hamsters were culled, and the trachea and lung were collected for further analysis.

We computed all p-values via bootstrap hypothesis testing, by subtracting from each hamster the mean of its group, resampling hamsters from each group with replacement 20000 times, and computing the fraction of samples in which the difference between the means was greater than the real difference. The presented p-values are after Bonferroni-correction for the four arms.

**Statistics and Reproducibility**. We performed the initial covid screen (Fig. 1) using two replicates (defined as separate sorted cells). We performed the screen validation (Fig. 2) using two replicates (defined as separate transfection experiments) for each tested siRNA at each concentration. We performed the live virus experiment (Fig. 3) using 3 replicates (defined as separate cell infection experiments) for each siRNA and performed statistical analysis using Dunnett's test. We performed the saturation mutagenesis analysis (Fig. 4) using 2,143 different siRNAs, testing three different mismatches per position. We performed the in-vivo experiments (Fig. 5) using six hamsters per group and performed statistical analysis using bootstrap hypothesis testing (subtracting from each hamster the mean of its group, resampling hamsters from each group with replacement 20000 times, and

computing the fraction of samples in which the difference between the means was greater than the real difference).

**Reporting summary**. Further information on research design is available in the Nature Portfolio Reporting Summary linked to this article.

## Data availability

Data from the sensor assay experiments, unprocessed data and metadata files is available upon request. The Source data for graphs and figures is included in supplementary data 1-5.

## Code availability

Code required to replicate the results is available from the authors upon request.

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

## Acknowledgements

This study was funded in part by the Bill & Melinda Gates Foundation and by the intramural program of the National Institutes of Health. The findings and conclusions contained within are those of the authors and do not necessarily reflect positions or policies of the Bill & Melinda Gates Foundation. This work was conducted in part under CRADA 2020-0597 between the VRC/NIAID and Eleven Therapeutics. Some of the work was funded by research grants to I.G. from Wellcome Trust, reference no. 207498/Z/17/Z and MRC/UKRI G2P-UK National Virology consortium, reference no. MR/W005611/1. I.G. is a Wellcome Trust Senior Fellow.

## Author contributions

O.Y. and Y.E. wrote this paper with input from G.B., D.B., O.W. and M.G. O.Y., Y.E. and O.W. drafted the figures and tables. O.Y., G.B., D.B., A.N. and I.F. and A.C.B carried on the in vitro experiments. R.I., M.H. and I.G.G carried on the live virus experiments and performed statistical analysis. L.M., T.J., C.C.H and M.G. carried on the in vivo study. Y.E., O.W. and R.R performed analyses of sequencing data and performed statistical analysis. Y.E., S.I., I.G., D.D., G.J.H contributed to discussion and study designs.

## Competing interests

O.Y., O.W, A.N., I.F., A.C.B, R.R., S.I., I.G. and Y.E. are Eleven Therapeutics employees. G.J.H is the Scientific co-founder and holds equity in Eleven Therapeutics. G.B. and D.B. hold equity in Eleven Therapeutics. I.G. was acting as an adviser in Eleven Therapeutics when this work was performed.

## Ethics

All in vivo studies were conducted according to NIH regulations and standards on the humane care and use of laboratory animals as well as the Animal Care and Use Committees of the NIH Vaccine Research Centre and BIOQUAL, Inc. (Rockville, Maryland). All studies were conducted at BIOQUAL, Inc.
