## [Peer Review File · Communications Biology]

REVIEWERS' COMMENTS:

Reviewer #2 (Remarks to the Author):

In the revised manuscript, the authors have reasonably addressed some critiques raised by the reviewer. While more experiments are needed for demonstration of the true potential of RNAi strategy for SARS-CoV2 prophylaxis, the genome-wide screen of siRNAs against the virus is interesting. Thus the reviewer would recommend publication of this work in Communications Biology.

Comment on authors' response to Reviewer #1:

I have reviewed the authors' responses to the critiques. I feel the authors have, overall, been responsive, and have made changes accordingly in the manuscript. Considering that this work should now be viewed as a methodology paper, the authors made a reasonable argument that the suggested experiments by the reviewers might be out of scope of this work. Regardless the efficacy of RNAi for COVID-19 prophylaxis, I think this paper may be valuable in terms of genome-wide screening of RNAi molecules for viral infection, and thus would recommend its publication in your journal.

October 29, 2022

Dear Dr. Zhijuan Qiu,

We would like to thank you for the opportunity to publish our manuscript in *Communications Biology*. The previous reviewers raised a series of important questions and suggestions, which we addressed by enhancing the clarity of the text, contextualizing the results of our work, and providing more data. Below is a point-by-point response to the reviewers' comments. To clarify, their comments can be broadly divided into three main topics:

The power of our *in-cellulo* genome-wide screen: the reviewers pointed out that both of our *in-cellulo* genome-wide screen and *in-silico* prediction algorithms aim to identify potent siRNAs. Thus, it is not a surprise that these two strategies are correlated with each other to some extent. However, the most challenging part of siRNA identification is selecting the most hyper-potent siRNAs out of the potent ones. Machine learning algorithms have difficulty accurately predicting the tails of a distribution, a phenomenon that is known as regression to the mean. Our strategy allows us to empirically test every possible RNAi trigger *in-cellulo* and get a readout in order to find these rare, hyper-potent triggers. In addition, our high throughput approach enabled us to conduct saturation mutagenesis screens and to assess the impact of each double mutant in the target site on the effect of the siRNA. To the best of our knowledge, this is the first time that such a screen is conducted. Moreover, there are no *in-silico* alternatives to predict such events, emphasizing the power of our strategy. Based on the reviewers' comments, we added to the manuscript more details to explain the value of our screen to the process we took to select the siRNA cocktail.

The range of *in-cellulo* models for SARS-CoV-2 infection: The reviewers also pointed whether VeroE6 models alone are sufficient to measure the effect of our siRNAs on SARS-CoV-2. We thank the reviewers for encouraging us to expand the work for additional models. Based on their comments, the manuscript now describes new data from SARS-CoV-2 infections in primary Air-liquid-interface culture.

Combination of other known drugs and additional *in vivo* studies: This study focuses on developing a novel siRNA screening pipeline which allows the fast identification of hyper-potent siRNAs against any target gene or genes and the ability of this pipeline to identify the significance of mismatches in each position. While we support this line of investigation, screening combinations of known drugs is not the main focus of this study and we suggest addressing this interesting question in future studies. As for the *in vivo* SARS-Cov2 studies, we inquired with the NIAID, which has been responsible for running these studies in a BSL3 facility. While we are planning more studies in the future, unfortunately at this point, there are no available time slots in the next few months. Due to the importance of our study, we do not wish to delay the publication any longer and suggest addressing the important comments of reviewers as caveats in the current study. Also, we found that a single *in-vivo* work is quite typical in multiple COVID related manuscript aimed for a discovery of a new molecule (e.g. Nature BioTechnology, VOL 39, June 2021, 717–726, Mol Ther. 2021 Jul 7, 2219–2226).

Thanks for your time and consideration,

Dr. Ohad Yogev

Director of Cellular Biology

Eleven Therapeutics

Reviewers' comments:

Reviewer #1:

Major points:

- Why are small molecule drugs not mentioned/used as complements to vaccines and antibodies, including protease & RT inhibitors (e.g. Paxlovid) and nucleotide prodrugs (e.g. Remdesivir)?

We thank the reviewer for this excellent point about the combination of small molecules and vaccines. Based on their request, we expanded the introduction to point out the availability of paxlovid and molnupiravir for treatment and the relatively underwhelming results of Pfizer to utilize paxlovid as a post-exposure prophylaxis, which suggest that it is not easy to translate successful treatments as prophylaxis.

- siRNAs have been developed against SARS before, in 2005, ref 18 (and ref 30-32). Have these siRNAs been compared to the current ones, and how did they fare in the assay? Why did those siRNAs (or similar siRNAs matching SARS-CoV-2) not translate during the pandemic despite having been tested in NHPs already?

Excellent point! In fact, testing in-silico these SAR-CoV siRNAs against SARS-CoV-2 genome was the first thing that we tried. Moreover, we had a pipeline that extracted all publicly available siRNAs sequences mentioned in pubmed against SAR-CoV and tried to find good matches to the SARS-CoV-2 genome. Unfortunately, out of 9422 sequences (from more than 3800 manuscripts) that were tested only 12 siRNAs had a perfect match to the SARS-CoV-2 genome. Individual study of each of these siRNAs did not lead us to believe that these sequences can be used for prophylactic treatment. To make the manuscript more complete, we added this point to the discussion.

- Fig 1C shows that there is an over 90% correlation between algorithmic prediction and screen outcome. Fig 1E and F show that screening outcomes partially validated, with IC50 values spanning more than 2 log scales. One way of interpreting this data is that existing algorithm might already be able to predict siRNA performance sufficiently. Did the authors try to validate 10 top candidates from these algorithmic predictions as comparison (later they do try predictions, but switch to other algorithms)?

Again, the reviewer raises an excellent point. As mentioned in the result section and in supplementary figure 3, we utilized 3 prediction algorithms, namely RNAXs, DSIR and OligoWalk, to identify potent target sites in the SARS-CoV-2 genome. This analysis returned 88 siRNAs that were classified as highly potent based on the consensus of the three algorithms. We tested these siRNAs individually and most of which did not generate a high repression in a reporter assay (see Supplemental Figure 4). This stands in a stark contrast to the top ten siRNA that were selected based on the Sens.AI screen that in most cases showed an IC50 of <20pM.

Indeed, these results could be perceived as surprising due to the high correlation in Figure 1C between the machine learning algorithm and the Sens.AI score. R^2 is calculated using the bulk of the data (i.e. mediocre to good siRNAs) where indeed there is a good correlation between the siRNAs. However, for discovery purposes, we are interested in the far right tail of the distribution (super potent siRNAs) where a weak correlation can appear between the strategies. This is not surprising – most machine learning algorithms for siRNA predictions are trained with only a few thousands of data points. Hyper potent siRNAs are needle in a haystack and appear <2% of the locations, meaning that the algorithm has only a handful of examples of hyper potent data points to learn from. We improved the text to clarify this point.

- Where in the manuscript are the sequences of the top candidates, and list of sequences of all shRNAs? Additionally, were the siRNAs made as 21-mers or as 22-mers, more reflecting the length of the shRNAs? 22-mers would (likely) increase the translatability of sequence features from shRNAs to siRNAs.

The reviewer is raising an excellent question regarding the effect of the siRNA length on its activity, which we are extensively testing. In this study, all siRNAs were 22-mers. We apologise for the oversight. All sequences of the siRNAs used in this study are now included in Table 1 and Supplementary Table 3. We thank the reviewer for this point that improves the reproducibility of our manuscript.

- The best validated siRNA (S8) had the lowest screen score from the set of 10 that were tested. When commenting on screening performance, this should be discussed. Also, what fraction of hits have a score of ~1.75 and above? Moreover, it might be good to validate low-scoring candidates (with screen scores around 0), to test false-negative rates of the assay.

The reviewer is raising an interesting question about the relationship between the screen and reporter assays, which use a short context and viral inhibition. While most of the tested siRNAs dramatically lowered the amount of viral RNA, the minority failed to do the same. Specifically, S8 and S10 showed weak viral inhibition despite very high potency in the reporter assay. Interestingly, unlike the other siRNA candidates, these two target the virus's negative strand, which is an intermediary in the replication process. Therefore, we hypothesised that targeting this intermediary RNA molecule likely would not interfere with viral replication. This point is now emphasised in the paper. We thank the reviewer for that point which improve the manuscript.

To answer the reviewer question regarding the hits above the score of 1.75, we included to the response the hit scoring histogram showing that 1.5% of our hits has a score higher than 1.75.

- The quantitative media assay should be explained in more detail and/or schematic in the main text/figures.

We added a description of the assay to the results and method sections. We thank the reviewer for raising this issue which makes the manuscript clearer.

- Fig 3C. It seems unclear why not a few more siRNAs from this set were validated here, e.g. more of the promisingly looking Nsp8 siRNAs?

The reviewer raise an important question about the target selection process. As shown in Supplementary Figure 3, compared to the siRNAs from the SensAI screen, most predicted siRNAs did not score well at the reporter assay. As live virus are much more complicated and require BSL3 facilities, we chose only the siRNAs which showed the highest inhibition in the reporter assay. Indeed, comparing to the SensAI siRNA, only Hel14 showed a similar viral inhibition.

- Why did the authors only use Vero E6 cells? It would be good to include validation in a human cell line system as well, particularly since the siRNA off-target analysis (computational) has been done against the human genome.

The reviewer raise an excellent point regarding the the selection of the cell model. While our initial validation was done in VeroE6, we completely agree that primary human cells are a better predictor of our siRNAs activity. To address this point we tested our lead cocktail in a live virus setting using an Air-Liquid Interface (ALI) culture (Figure 3E). We thank the reviewer for this comment, which improved our manuscript.

- Fig 4C seems to indicate that position 14 mutations/mismatches have some of the strongest effects on S5 potency (beyond positions 3/4/5). Yet, earlier the authors commented: "Interestingly, S5 is tolerant and shows efficacy against the Delta strain, despite the fact that it has a mutation in position 14 of its target site." This observation somewhat questions the predictability of this assay approach with regards to live virus.

Indeed, position 14 has a strong median efficacy on S5 potency, but it also has the widest confidence interval across all positions. Hence, there is inherent variability in mismatch tolerance in this position. We believe that the discrepancy between the median effect size (as measured by the reporter assay) and the results shown in the live virus assay simply reflects this variability.

Minor points:

- The concept of “original antigenic sin” should be explained in the text.

We thank the reviewer for the comment and added an explanation to the text.

- Are the siRNAs supposed to be a “mitigation of transmission” or a “prophylaxis” for the patient?

The function of the siRNAs as prophylactic treatment or transmission mitigation and the relationship between the two is fascinating. In this study we focused on testing prophylaxis, however based on the accumulating data from the vaccine, we hypothesized that reducing infection will result in reduced transmission.

- Use of sgRNA (for sub-genomic RNA), without explanation, is generally confusing, given the much more prevalent use in the context of CRISPR.

We thank the reviewer for this comment and edited the text to replace the phrase to subgenomic.

- Sup Fig 1. Resolution is insufficient (also true for other figs), size markers and/or scale bars should be included, legend for all abbreviations/proteins should be included.

This is a fair point. We have all figures as vectors and will provide high resolution figure to the final version of the manuscript as separate files.

- Sup Table 1. Check references.

- Fig 1. Spelling (modulation, precision...)

We amended the figure to address this comment. We thank the reviewer for spotting these spelling errors and amended the figure accordingly.

- S5 is mentioned in the text to be selected because it matches SARS-CoV as well. However, Table 1 says otherwise. S4 seems to be in that group according to Table 1.

We apologise for the oversight S5 has an imperfect match to SARS-CoV-1 (alignment of 19 nucleotides out of 22, with one mismatched among these 19 nucleotides). We thank the reviewer for spotting this mistake and amended the text accordingly.

- The FDA only gave an emergency use authorization (EUA) – did not approve – Bamlanivimab, and that authorization has been revoked. The manuscript should be updated.

We thank the reviewer for this comment and edited the manuscript accordingly.

- Fig 3. Where is the depletion data from the replicon system shown?

We thank the reviewer for this comment and added the reference to our manuscript showing that information (Berkyurek et al).

- Fig 3D and main text. Why were S5/S3 and S5/Hel14 prioritized over e.g. S5/S9?

The reviewer raise an excellent point regarding the target selection process. While S5/S9 worked well as a combination, S9 showed lower inhibition in the TCID₅₀ assay (Figure 3B). As multiple siRNA combinations achieved similar inhibition, we took into consideration all the data we had for each one of them as single treatment. Moreover, Hel14 was also prioritised due its high sequence conservation of its target gene its and essential role in viral replication.

- Fig 5. Statistics are used to compare the Treatment siRNA vs. Bamlanivimab. However, this is a bit apple vs. oranges as one is I.N. the other I.P., and the siRNA has 3 treatments before exposure, while the control only has 1.

The reviewer is raising an important point regarding the chosen positive control for our *in vivo* experiment. We agree with the reviewer that a positive control, which is administrated using the same method will be better. However, up to date, there are no available drugs or siRNAs that can be administrated using the same method and use as a positive control. As we still wanted to compare our siRNAs to a known positive control, we chose the best next alternative and administrated bamlanivimab as a one. We thank the reviewer for raising this point and it is now explained better in the manuscript.

Reviewer #2:

1. In Figure 2D, what are the criteria for manually selecting those ten candidates for further studies, given that two of the ten candidates showed much higher IC50 values than the rest? How representative are the selected candidates?

The reviewer raise an excellent question about the target selection process we used to test the ten candidates. Five out of the ten were selected mainly based on their average screen score while the other five were selected based on their screen score and conservation in both SARS-CoV and SARS-CoV-2. This process is explained in more details in the manuscript.

2. From Figure 3D, why not selecting the combination of S2/S5 or S5/S9 for further studies, with their viral load repression comparable or even better than that of the combination of S3/S5 or S5/H114?

The reviewer raise an excellent point regarding the target selection process. While S2/S5 and S5/S9 worked well as a combination, S2 and S9 showed lower inhibition in the TCID₅₀ assay (Figure 3B). As multiple siRNA combinations achieved similar inhibition, we took into consideration all the data we had for each one of them as single treatment. Moreover, Hel14 was also prioritised due its high sequence conservation of its target gene its and essential role in viral replication.

3. The prophylaxis efficacy shown in Figure 5 and Figure S7 may not strongly support the translational potential of the selected cocktails. The siRNA treatment only reduced viral load by ~10-fold (or less) and was much less effective compared to bamlanivimab. More characterization such as lung pathology should be further provided. More importantly, how could the efficacy be improved -- higher doses, different combinations, more siRNAs in the combination?

We agree with the reviewer notion that further optimisation is required to fully translate our *in vivo* data. This mainly includes optimisation of the siRNA delivery to the target cells and the administration dose and regimen. Nevertheless, the impact of this study does not depend solely on the *in vivo* data. Our work describes a discovery pipeline that can in a short period of time identify and validate siRNAs against any new sequence and predict their sensitivity for mutation. This pipeline is an extremely important tool that can be used in any future pandemic. Our *in vitro* and *in vivo* results support this notion showing that the identified siRNAs are highly active. While we plan to further optimise our *in vivo* studies to move forward with our siRNAs as anti-COVID drug, we believe that these are out of the scope of this manuscript. This point is emphasised in the end of the discussion.

4. Figure S7 showed lower viral load for modified siRNA administered by intranasal or nebulizer on day 5, but there was no effect on body weight change. This needs to be further clarified. There were no data to support 'slower pharmacokinetics' or 'less tolerable profile' either.

We agree with the reviewer that the *in vivo* administration of the siRNA still needs to be optimised. As part of it, we need to understand the pharmacokinetics when using different administration routes. Nevertheless, we believe that this optimisation is beyond the scope of this manuscript. To make this point clearer, we discuss these limitations in the discussion section.

5. The durability of prophylactic effect needs to be tested.

The durability of the prophylactic effect is an excellent question which needs to be addressed. However, as *in vivo* BSL3 studies are complexed, and these type of studies should be done once the delivery of the siRNAs is optimised. It is one of the studies we are planning for the future, but it is beyond the scope of this manuscript.

6. The therapeutic effect of siRNA cocktails after viral infection also needs to be examined.

Again, an excellent question which we aim to address in future studies. Post-infection studies in BSL3 are extremely complex and we believe that this question could be only addressed once the siRNA administration is fully optimised.

7. How accurate could it be for the exclusion by the in silico filters to exclude the seed region match human transcripts since it would be caused the side effect?

This is an excellent point and we agree that *in silico* filters by themselves cannot eliminate the possibility of off targets. As we progress with our drug development program, we aim to perform a detailed examination aiming

to identify human targets of our siRNAs and their potential to cause toxicity. However, we believe that this study is out of the scope of the current manuscript.